# Pooling and Attention: What are Effective Designs for LLM-based Embedding Models?

## Abstract

The significant advancements of Large Language Models (LLMs) in generative tasks have led to a growing body of work exploring LLM-based embedding models. While these models, employing different pooling and attention strategies, have achieved state-of-the-art performance on public embedding benchmarks, questions still arise about what constitutes an effective design for LLM-based embedding models. However, these models are often trained on different datasets, using different LLM base models or training settings. Moreover, evaluations on public embedding benchmarks often fail to report statistical significance, making it difficult to determine which designs truly contribute to final performance. This complicates the process for practitioners seeking optimal training recipes for LLM-based embedding models. In this study, we conduct a large-scale experiment by training a series of LLM-based embedding models using the same training data and base model but differing in their pooling and attention strategies. The results show that there is no one-size-fits-all solution: while bidirectional attention and an additional trainable pooling layer outperform in text similarity and information retrieval tasks, they do not significantly surpass simpler designs like EOS-last token pooling and default causal attention in clustering and classification tasks. Furthermore, we propose a new pooling strategy, Multi-Layers Trainable Pooling, which transforms the outputs of all hidden layers, rather than just the last layer, using a cross-attention network. This method proves to be statistically superior in text similarity and retrieval tasks compared to existing pooling methods. Overall, this paper sheds light on effective training strategies for LLM-based embedding models.

## 1 Introduction

A text embedding is a high-dimensional representation that captures the semantic information of text and is crucial for many tasks, such as information retrieval and semantic textual similarity. For example, text embedding models, which convert input text into embeddings, are essential components in semantic search and retrieval-augmented generation (RAG) retrieval-augmented generation systems (RAGs) (Gao et al., 2023; Hu and Lu, 2024). Companies such as OpenAI and Cohere provide embeddings as services via APIs.

Previous studies primarily employ encoder-only models, such as BERT (Devlin et al., 2019) and Sentence-BERT (Reimers, 2019), as embedding models. Recently, with the significant advancements in LLMs, the community has started to explore using LLMs as base models, fine-tuning them accordingly to serve as embedding models (Wang et al., 2023; Springer et al., 2024; Lee et al., 2024). On embedding model benchmarks such as Massive Text Embedding Benchmark (MTEB) (Muennighoff et al., 2022), LLM-based embedding models show promising performance and dominate the leaderboard compared to previous encoder-only models.

**Pooling** and **attention** are two main designs involved in converting an LLM into an embedding model. Pooling strategies are used to obtain a fixed-size dense vector representing the input sequence. For example, E5-mistral-7b-instruct (Wang et al., 2023) and SRF-Embedding-Mistral (Meng et al., 2024) use EOS-last token pooling as their pooling strategy. NV-Embed (Lee et al., 2024) uses a trainable pooling layer to obtain the final embeddings. The attention strategy constrains the direction in which tokens can attend to others. By default, an LLM is pre-trained with

a causal attention mask (Radford et al., 2018), meaning that a token can only attend to preceding tokens. However, several recent works have highlighted the potential limitations of causal attention for representation learning and propose that an LLM-based embedding model should allow bidirectional attention so that every token in the sequence can attend to every other token (BehnamGhader et al., 2024; Lee et al., 2024).

Table 1: State-of-the-art LLM-based embedding models. They vary in pooling and attention strategies. The "Score" column represents the performance on the MTEB benchmark (Muennighoff et al., 2022) as reported on the Hugging Face MTEB Leaderboard [1]

| Embedding Model | Base Model | Pooling | Attention | Training Data Size | Score |
|---|---|---|---|---|---|
| e5-mistral-7b-instruct | Mistral-7B-v0.1 | EOS-Last token pool | Causal | 1.8M | 66.63 |
| SFR-Embedding-Mistral | Mistral-7B-v0.1 | EOS-Last token pool | Causal | Not Specific | 67.56 |
| GritLM-7B | Mistral-7B-v0.1 | Mean pool | Bidirectional | Not Specific | 66.76 |
| LLM2Vec-Mistral-supervised | Mistral-7B-v0.1 | Mean pool | Bidirectional | 1.5M | 64.80 |
| LLM2Vec-Llama-2-supervised | Llama-2-7b | Mean pool | Bidirectional | 1.5M | 64.14 |
| NV-Embed-v1 | Mistral-7B-v0.1 | Trainable pooling layer | Bidirectional | 1.1M | 69.32 |

Table 1 lists state-of-the-art LLM-based embedding models. Some perform better than others. This raises the question: what makes an embedding model perform better? Is it the higher quality of the fine-tuning dataset, the greater capability of the base LLM, or the use of different pooling and attention strategies that makes the embedding model more effective? Unfortunately, most existing LLM-based embedding models are trained using different datasets with different base models, making it difficult to draw conclusions regarding the contribution of each design choice.

In this paper, we conduct large-scale experiments to empirically evaluate pooling and attention strategies for LLM-based embedding models. To ensure a fair comparison between different strategies, we fine-tune the same base LLM models (Mistral-7B and Qwen2-0.5B) using different combinations of pooling and attention strategies commonly employed in existing models. We conduct statistical testing to rigorously compare the performance of these models. Interestingly, we find that there is no one-size-fits-all solution. For example, LLMs with bidirectional attention and an additional trainable pooling layer demonstrate superior performance in semantic textual similarity (STS) and information retrieval tasks but underperform in clustering and classification tasks.

In addition to empirically testing existing pooling strategies, we also propose a new pooling strategy, Multi-Layers Trainable Pooling, which leverages LLM hidden states across multiple internal layers and transforms them using a trainable network. This strategy is motivated by the observation that different internal layers in LLM may encode orthogonal information that is not captured in the final layer but could be relevant for certain downstream tasks. Empirical experiments show that this new pooling strategy, which pools information from multiple layers, outperforms existing methods that use only the last layer. Overall, we hope that these large-scale training experiments and the proposed pooling strategy can collectively enhance the community's efforts to improve LLM-based embedding model performance. We will release the implementation of the proposed pooling method and the series of fine-tuned embedding models for replication.

## 2 COMMONLY USED POOLING AND ATTENTION STRATEGIES

In this section, we briefly review different pooling and attention strategies that are commonly used in existing LLM-based embedding models.

### 2.1 POOLING STRATEGY

A pooling strategy focuses on obtaining a fixed-size embedding from the LLM hidden states for an input sequence. We denote the LLM hidden states as a matrix $\mathbf{H} \in \mathbb{R}^{l \times n \times d}$, where $l$ is the hidden layer, $n$ is the sequence length, and $d$ is the hidden size. Three commonly used pooling strategies for matrix $\mathbf{H}$ are widely used.

**EOS-Last Token Pooling:** $\mathbf{h}_{\text{eos}} = \mathbf{H}_{[-1,n,:]}$ Since the next-word prediction is often the training objective for LLM, the last token of the whole input sequence, therefore, captures all the information

---

[1]https://huggingface.co/spaces/mteb/leaderboard

of the sequence. Thus, many existing works, such as OpenAI's cpt-text model (Neelakantan et al., 2022) and E5-mistral-7b-instruct (Wang et al., 2023), append a special End-of-Sequence (EOS) token to the input and use the last layer's hidden states of the EOS token as the text embedding.

**Mean Pooling:** $\mathbf{h}_{\text{mean}} = \frac{1}{n} \sum_{i=1}^{n} \mathbf{H}_{[-1,i,:]}$ Here, $\mathbf{h}_{\text{mean}}$ denotes the embedding obtained by averaging the last layer hidden states of all tokens in the sequence. Some existing works, such as GritLM-7B (Muennighoff et al., 2024) and LLM2Vec (BehnamGhader et al., 2024), employ this pooling strategy on the LLM-based embedding model.

**Trainable Pooling Layer:** Instead of directly using the LLM hidden states as the input embedding, NV-embed (Lee et al., 2024) pioneers a novel method using an additional trainable pooling layer to convert LLM's last layer's hidden states into a semantic latent space: $\mathbf{h}_{\text{pool}} = \mathbf{M}(\mathbf{H}_{[-1,:,:]})$, where $\mathbf{H}_{[-1,:,:]}$ is the last hidden state from LLM and $\mathbf{M}$ is a trainable network.

## 2.2 ATTENTION STRATEGY

LLMs are mostly pre-trained using a causal attention mask (Radford et al., 2018), a unidirectional attention mechanism that allows the current token to only attend to preceding tokens. However, several recent studies have demonstrated the limitations of unidirectional attention and have adapted the attention mask of the LLM to be bidirectional during the fine-tuning process, allowing each token to access the bidirectional context in the sequence (BehnamGhader et al., 2024; Lee et al., 2024; Springer et al., 2024). Subsequently, we denote these two strategies as **Causal** attention and **Bidirectional** attention.

## 2.3 FINE-TUNING LLMS AS EMBEDDING MODELS

While existing LLM-based embedding models differ in their pooling and attention strategies, the training process is largely similar. To fine-tune an LLM as an embedding model, the contrastive learning method is often used (Henderson et al., 2017; Wang et al., 2023; Lee et al., 2024). In short, contrastive learning encourages the embedding of a focal example to be similar to that of a positive example while being distant from its negative example, thus enabling a base LLM to adapt to embedding-related tasks. All the works listed in Table 1 use contrastive learning to fine-tune the base LLM. However, they use different training data, so the final performance may be confounded by the dataset.

## 3 MULTI-LAYERS TRAINABLE POOLING: A NEW POOLING STRATEGY THAT OBTAINS EMBEDDING FROM MULTIPLE LAYERS

As shown in Table 1, state-of-the-art LLM-based embedding models use pooling strategies that obtain embeddings from the last layer of the LLM's hidden states, regardless of whether they employ EOS-last token pooling, mean pooling, or trainable pooling. But can we achieve better results by using hidden states from other layers? Prior work has shown that different layers of language models, such as encoder-only BERT or decoder-only LLMs, encode different semantic information (Clark, 2019; Oh et al., 2022; Ju et al., 2024). Therefore, we hypothesize that the other layers may contain relevant information that complements the last layer. In this section, we first verify that each layer's hidden states encode distinct information and that the intermediate layers may also contain relevant information beneficial to downstream tasks. We then propose a new pooling method, termed **Multi-Layers Trainable Pooling**, which consists of a trainable pooling layer that uses hidden states from all layers.

### 3.1 LAST LAYER VS. OTHER LAYERS

We conduct two experiments to compare the hidden states of the last layer with those of other layers.

**Experiment 1: Different Hidden State Layers Encode Distinct Aspects.** In this experiment, we measure the correlation of hidden states across different layers. Specifically, we select two datasets, HotpotQA (Yang et al., 2018) and MS MARCO (Bajaj et al., 2016), and append an EOS token to the input sequences. We then pass the input sequences through two LLMs, Mistral-7B-v0.1 (Jiang et al., 2023) and Llama3-8B (Dubey et al., 2024), and obtain the hidden states of the EOS token from

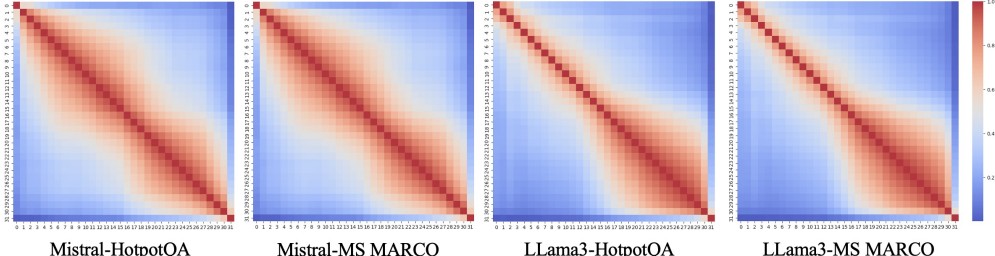

Mistral-HotpotQA          Mistral-MS MARCO          LLama3-HotpotQA          LLama3-MS MARCO

Figure 1: The correlation heatmap of EOS token hidden states across different layers. The two figures on the left are measured on Mistral-7B-v0.1 using the HotpotQA and MS MARCO datasets, while the two figures on the right are measured on Meta-Llama-3-8B. Areas shaded in blue indicate low correlation, while areas shaded in red denote high correlation. The horizontal axis represents the layer index ranging from 0 to 31, while the vertical axis represents the layer index ranging from 31 to 0.

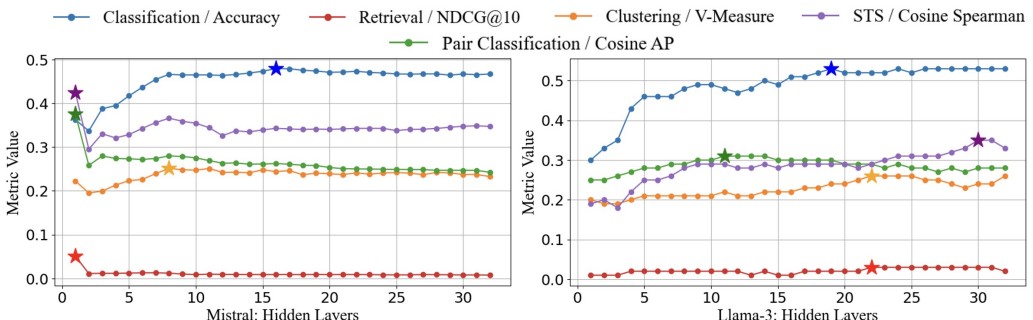

Figure 2: Performance of different hidden layers from Mistral-7B-v0.1 (left) and Llama3-8B (right) on the MTEB benchmark. The highest score is marked with a star. The X-axis represents the layer index ranging from 0 to 31, and the Y-axis represents the performance score.

different layers. Note that neither of the base LLMs has been fine-tuned as an embedding model. For each input sequence, we measure the Spearman's correlation coefficient between the hidden states of different layers. The layer-wise correlation heatmaps are shown in Figure 1.

The results clearly indicate that the embeddings from adjacent layers are more correlated than those from layers further apart. More importantly, the findings reveal that embeddings from different layers, particularly those that are not adjacent, are vastly different and may encode distinct aspects. Although the LLMs are base models and have not been fine-tuned as embedding models, this observation suggests that the hidden states learned by different layers within LLMs are not entirely the same, indicating a variation in the information captured across the layers.

**Experiment 2: Other Layers' Hidden States May Also Be Useful for Downstream Tasks.** In this experiment, we assess the representation capability of the hidden states in different layers on downstream tasks. Specifically, we evaluate the downstream task performance of EOS token embedding of each layer, using the Mistral-7B-v0.1 and Llama3-8B models on the MTEB benchmark(Muennighoff et al., 2022). The results are shown in Figure 2.

Interestingly, and perhaps surprisingly, the hidden states of the last layer do not perform the best across MTEB tasks. For Mistral-7B-v0.1, hidden states from earlier layers capture more semantic information than those from later layers. The performance gap between the last layer and the best-performing layer is 0.08 for the STS task and 0.04 for the retrieval task. For the Llama3 model, the middle layers appear to be more effective at encoding semantic meaning.

Although the behavior of the hidden states in intermediate layers will change in fine-tuned embedding models, the key takeaway from these experiments is that the hidden states of other layers may

encode information that complements that of the last layer and could be useful for downstream tasks. Thus, relying solely on the last layer's hidden state in the pooling strategy may not be optimal.

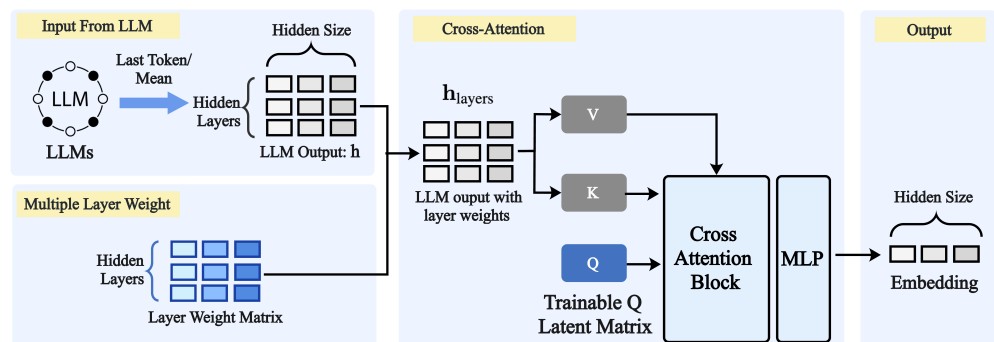

Figure 3: The proposed pooling method: Multi-Layers Trainable Pooling. It combines the EOS token hidden states from all layers in the LLM and transforms them into the final embedding using a cross-attention network.

## 3.2 MULTI-LAYERS TRAINABLE POOLING

Motivated by the above findings, we propose a new pooling strategy termed Multi-Layers Trainable Pooling, which utilizes an additional trainable layer to capture semantic information from *all layers* in an LLM. The method is shown in Figure 3. Let $\mathbf{H} \in \mathbb{R}^{l \times n \times d}$ represent the LLM hidden state matrix of an input sequence, where $l$ is the number of hidden layers, $n$ is the sequence length, and $d$ is the hidden size. The high-level idea of Multi-Layers Trainable Pooling is to introduce a trainable layer that learns to pool hidden states from different layers. Specifically, there are three main components in this pooling method:

**Input: LLM Hidden States Across All Layers.** The first step involves selecting the LLM output $\mathbf{H}$ as the input for the pooling operation. For causal attention LLMs, we use the EOS token hidden states of all LLM layers, denoted as $\mathbf{h}_{\text{causal}} = \mathbf{H}_{[:,-1,:]}$, as the input to the subsequent trainable pooling layer. This approach is chosen because, in causal attention LLMs, earlier tokens may introduce bias, subsequently affecting the final embedding (Springer et al., 2024). For bidirectional attention LLMs, we consider the mean embedding across the token length dimension, expressed as $\mathbf{h}_{\text{bi-directional}} = \frac{1}{n} \sum_{i=1}^{n} \mathbf{H}_{[:,i,:]}$. The resulting vector serves as the input to the subsequent trainable pooling layer.

**Layer Weight Matrix.** Different layers may have varying importance to the final embedding depending on the task. To account for this, we introduce a trainable layer weights matrix that captures the significance of each layer. Specifically, we combine the LLM output with a trainable layer weight matrix, $\mathbf{W} \in \mathbb{R}^{l \times d}$. The combined layer matrix is computed as $\mathbf{h}_{\text{layers}} = \mathbf{h} + \mathbf{W}$, where $\mathbf{h}$ is the input (either $\mathbf{h}_{\text{causal}}$ or $\mathbf{h}_{\text{bi-directional}}$).

**Cross Attention Matrix.** The combined layer matrix $\mathbf{h}_{\text{layers}}$ is then multiplied by the parameter matrices $W_K$ and $W_V$ and produce the key matrix $\mathbf{K}$ and value matrix $\mathbf{V}$ accordingly. The $\mathbf{K}$ and $\mathbf{V}$ matrix is then combined with a trainable query matrix $\mathbf{Q} \in \mathbb{R}^{r \times d'}$, with $d'$ being the inner dimension of the cross-attention block and $r$ being the number of latent dimensions which is the same with LLM hidden dimension. Note that the $\mathbf{K}$ and $\mathbf{V}$ are both derived from the linear transformation of input $\mathbf{h}_{\text{layers}}$, while $\mathbf{Q}$ is a trainable parameter matrix. This cross-attention network is similar to the method used in Flamingo (Alayrac et al., 2022), which maps varying-size video frames to fixed-size visual embeddings, similar to obtaining embeddings from multiple layers.

The cross-attention network computes attention between fixed, trainable queries and keys/values derived from the input to capture and encode the most relevant information from $\mathbf{h}_{\text{layers}}$ into a semantic latent space. The output of this cross-attention network is then passed through a multi-layer perceptron (MLP) to produce the final output embedding. Using a trainable query matrix ($\mathbf{Q}$) instead of deriving it directly from the input data ($\mathbf{h}_{\text{layers}}$) allows the cross-attention mechanism to more effectively filter semantic information from multi-layer hidden states through the trainable $\mathbf{Q}$.

Compared to NV-embed (Lee et al., 2024), which pioneered the use of a trainable pooling layer, our proposed pooling method introduces three key innovations. First, we utilize hidden states from all layers rather than just the last layer. Second, we include a trainable layer weight matrix to account for the varying importance of different layers across tasks. Third, while NV-embed transforms the last layer's hidden states to the query matrix $\mathbf{Q}$, our method combines the hidden state matrix with the layer weight matrix and transforms it into key matrix $\mathbf{K}$ and value matrix $\mathbf{V}$.

## 4  POOLING AND ATTENTION EXPERIMENTS

As shown in Table 1, state-of-the-art embedding models are often trained using different pooling and attention strategies. However, two factors obscure our understanding of the most effective designs for LLM-based embedding models. First, they are trained using different datasets, which is a key confounding factor that significantly influences final performance on the MTEB benchmarks. Second, they primarily report results without statistical testing, making it unclear whether the observed performance improvements are statistically meaningful. To address these, we aim to empirically assess the effectiveness of different pooling and attention strategies through a fair comparison using the same dataset and training protocol.

### 4.1  POOLING AND ATTENTION COMBINATIONS

In the experiment, we consider the following five design combinations:

**Model 1:** EOS-Last Token Pooling + Causal Attention

**Model 2:** Last-Layer Trainable Pooling + Causal Attention

**Model 3:** Multi-Layers Trainable Pooling+ Causal Attention

**Model 4:** Last-Layer Trainable Pooling + Bidirectional Attention

**Model 5:** Multi-Layers Trainable Pooling+ Bidirectional attention

These five combinations allow us to conduct pairwise comparisons between different pooling and attention strategies by controlling for other potential confounding factors. For example, by comparing Model 1, Model 2, and Model 3, we can assess the effectiveness of different pooling strategies under the same attention method. Similarly, by comparing Model 3 and Model 5, we can evaluate how different attention methods affect embedding performance when using a multi-layer trainable pooling strategy.

**Why Mean Pooling Is Not Considered in Our Setting?** First, prior work has demonstrated that employing mean pooling in a causal attention LLM-based embedding model introduces a bias towards the earlier tokens (Springer et al., 2024), leading to poor performance (Wang et al., 2023; BehnamGhader et al., 2024). Second, for LLMs that use bidirectional attention, NV-embed (Lee et al., 2024) has shown that an additional trainable pooling layer can outperform mean pooling. Thus, mean pooling does not appear to be a viable choice for either attention strategy. To keep our experiment manageable in scope, we have therefore excluded mean pooling from consideration.

**Why EOS-Last Token Pooling + Bidirectional Attention Is Not Considered in Our Setting?** Intuitively, when bidirectional attention is used, the EOS-Last token is no longer meaningful as the input embedding. In fact, existing LLM-based embedding models that use bidirectional attention typically employ pooling techniques such as mean pooling or trainable pooling layers (Lee et al., 2024; BehnamGhader et al., 2024; Muennighoff et al., 2024). Therefore, in our experiment, we do not include this combination of pooling and attention.

### 4.2  EXPERIMENTAL DETAILS

**Base LLM.** We use the Mistral-7B-v0.1 model as the base LLM. We choose Mistral-7B because it is widely regarded as one of the best open-source LLM models for embeddings and is commonly used in state-of-the-art embedding models, as shown in Table 1.

**Training data.** We use publicly available datasets that are commonly utilized for embedding model fine-tuning to train Model 1 to Model 5. Since the models are trained on the same dataset but differ

in either pooling or attention strategy, this approach allows us to isolate the impact of these strategies and ensures a fair comparison between them. The size of the training dataset is 1.4 million, which is about the same scale as existing works illustrated in Table 1. An additional EOS token is appended to the end of each training example. Following e5-mistral-7b-instruct (Wang et al., 2023), we also include instructions in the query to describe the task. Table 6 in the Appendix lists the training datasets and associated instructions used in this study.

**Contrastive Learning.** For contrastive learning, we follow the standard training pipeline that each query is paired with one positive and one hard negative example (Wang et al., 2023). The positive example is provided by the datasets, while the hard negative example is mined by a trained SentenceTransformer [2]. During training, we utilize in-batch negatives, where the negative examples for a given query are sourced from the other queries within the same batch.

**Training Setting.** We use Low-Rank Adaptation (LoRA) (Hu et al., 2021) with a LoRA rank of 16 to finetune the model for downstream embedding tasks using contrastive learning loss. The learning rate is 1e-5, and the training batch size is 2,048. The max training step is 1,000, which is aligned with existing works (Wang et al., 2023; BehnamGhader et al., 2024).

**Trainable Pooling Layers Setting.** The Last-Layer Trainable Pooling employs query and cross-attention dimensions consistent with the hidden size of the Large Language Model (LLM), which is 4,096 for Mistral-7B. The multi-head cross-attention mechanism utilizes 32 heads, each containing 2,048 channels. This setting aims to reproduce a module similar to NV-embed (Lee et al., 2024). The proposed Multi-Layers Trainable Pooling shares the same basic module parameters as the Last-Layer Trainable Pooling, except for the trainable layer weight matrix. Specifically, the trainable layer weight is $\mathbf{W} \in \mathbb{R}^{32 \times 4096}$.

**Evaluation.** We evaluate all five fine-tuned models on the MTEB Benchmark (Muennighoff et al., 2022) encompassing 15 retrieval datasets, 4 reranking datasets, 12 classification datasets, 11 clustering datasets, 3 pair classification datasets, 10 semantic textual similarity datasets, and 1 summarization dataset. Table 7 in the Appendix lists all the evaluation tasks and instructions.

**Wilcoxon Signed Rank Test.** While the MTEB benchmark is commonly used in prior LLM-based embedding models, it is unfortunate that statistical significance is not commonly reported. As a result, it remains unclear whether the improvement of a specific design, such as bidirectional attention, is statistically meaningful. The MTEB benchmark comprises seven tasks, with each task containing a different number of datasets. To ensure statistical rigor and accurately assess model performance on each task, we conduct a Wilcoxon Signed Rank Test within each task to determine the statistical significance of the experimental results. Tasks with four or fewer datasets—including reranking, pair classification, and summarization—are excluded from this test due to the extremely small sample size, which limits the statistical power of the analysis. Therefore, we employ the Wilcoxon Signed Rank Test on the Retrieval, STS, Classification, and Clustering tasks. **We consider the comparison to be significant when the p-value is less than 0.05.**.

## 5 EMPIRICAL ANALYSIS

After fine-tuning the Mistral-7B with various configurations (Model 1 $\sim$ Model 5) and testing on the MTEB benchmark, we provide an empirical analysis of the effectiveness of different pooling and attention strategies.

### 5.1 WHAT IS THE OPTIMAL POOLING STRATEGY?

In this section, we compare EOS token pooling, Last-Layer Trainable Pooling, and proposed Multi-Layers Trainable PoolingThe Last-Layer Trainable Pooling is similar to Multi-Layers Trainable Pooling but the input contains only the last hidden states and without trainable layer weights. The results are illustrated in the Table 2.

**Finding (1): An additional trainable pooling layer is preferable only for STS tasks, but not other tasks, when causal attention is used.** For the STS task, the performance of EOS-Last token pooling is statistically lower than using a trainable pooling layer (regardless of whether using

---

[2]https://huggingface.co/sentence-transformers/all-MiniLM-L6-v2

Table 2: Comparison of pooling strategies on the MTEB benchmark. The score represents the average score across datasets within each task. The row in  gray  is the baseline for comparison, and the pairwise significant results are marked with asterisks* with a p-value less than 0.05.

| Combination | Pooling | STS | Clas. | Retr. | Clus. |
|---|---|---|---|---|---|
| **Casual Attention** | | | | | |
| Model 1 | EOS-Last Token Pooling | 0.8302 | 0.7244 | 0.5394 | 0.4503 |
| Model 2 | Last-Layer Trainable Pooling | **+0.0129*** | -0.0035 | +0.0102 | -0.0076 |
| Model 3 | Multi-Layers Trainable Pooling | **+0.0118*** | -0.0033 | +0.0135 | -0.0017 |
| Model 2 | Last-Layer Trainable Pooling | 0.8431 | 0.7209 | 0.5496 | 0.4427 |
| Model 3 | Multi-Layers Trainable Pooling | -0.0011 | +0.0002 | +0.0033 | +0.0059 |
| **Bi-directional Attention** | | | | | |
| Model 4 | Last-Layer Trainable Pooling | 0.8397 | 0.6761 | 0.5607 | 0.4010 |
| Model 5 | Multi-Layers Trainable Pooling | **+0.0071*** | **+0.034*** | **+0.0013*** | **+0.0247*** |

Last-Layer Trainable Pooling or Multi-Layers Trainable Pooling). As shown in Table 2, the Last-Layer Trainable Pooling and Multi-Layers Trainable Pooling both achieve significant improvement (+0.0129 and +0.018 respectively) compared to EOS token pooling. However, both trainable pooling methods fail to pass the significance test for the classification, retrieval, and clustering tasks.

**Finding (2): Using Multi-Layers trainable pooling is more effective than Last-Layer trainable pooling when bidirectional attention is used.** As shown in the bottom rows of Table 2, Multi-Layers Trainable Pooling significantly outperforms Last-Layer Trainable Pooling across all four tasks (STS, Classification, Retrieval, and Clustering). However, no significant results are observed in the causal attention LLM setting across four tasks using the Wilcoxon Signed Rank Test, suggesting that the benefits of multi-layer pooling are context-dependent.

## 5.2 WHAT IS THE OPTIMAL ATTENTION STRATEGY?

Table 3: Comparison of attention strategies on the MTEB benchmark. The score represents the average score across datasets within each task. The row in  gray  is the baseline for comparison, and the pairwise significant results are marked with asterisks*.

| Combination | Attention | STS | Clas. | Retr. | Clus. |
|---|---|---|---|---|---|
| **Last-Layer Trainable Pooling** | | | | | |
| Model 2 | Casual Attention | 0.8431 | 0.7209 | 0.5496 | 0.4427 |
| Model 4 | Bi-directional Attention | -0.0034 | -0.0448 | **+0.0111*** | **-0.0417*** |
| **Multi-Layers Trainable Pooling** | | | | | |
| Model 3 | Casual Attention | 0.8420 | 0.7211 | 0.5529 | 0.4486 |
| Model 5 | Bi-directional Attention | +0.0048 | -0.011 | **+0.0091*** | **-0.0229*** |

**Finding (3): Bi-directional attention is better at retrieval task but worse at clustering task.** The data from Table 3 demonstrates that bi-directional attention masks consistently improve performance in the retrieval tasks, regardless of the pooling strategy employed, although the absolute improvement on the retrieval task is trivial. In contrast, the same configuration leads to diminished performance in the clustering tasks, as indicated by the negative deltas in scores (-0.0417 for Last-Layer Trainable Pooling and -0.0229 for Multi-Layers Trainable Pooling). These divergences suggest that the bidirectional attention strategy enhances the model's capacity to consider the context from both directions, proving beneficial for retrieving relevant information. However, this increased context may also introduce noise, which can hinder effective clustering.

## 5.3 What is the Optimal Pooling and Attention Design?

The performance of Model 1 ∼ Model 5 is presented in Table 4.

Table 4: Comparison of different pooling and attention combinations on MTEB benchmark. **Mistral-7B-v0.1** is the base LLM. The row in gray is the baseline for comparison, and the pairwise significant results are marked with asterisks*.

| Combination | Pooling | Attention | STS | Clas. | Retr. | Clus. |
|---|---|---|---|---|---|---|
| Model 1 | EOS-Last Token Pooling | Casual | 0.8302 | 0.7244 | 0.5394 | 0.4503 |
| Model 2 | Last-Layer Trainable Pooling | Casual | **+0.0129*** | -0.0035 | +0.0102 | -0.0076 |
| Model 3 | Multi-Layers Trainable Pooling | Casual | **+0.0118*** | -0.0033 | +0.0135 | -0.0017 |
| Model 4 | Last-Layer Trainable Pooling | Bi-directional | **+0.0095*** | -0.0483 | +0.0213 | **-0.0493*** |
| Model 5 | Multi-Layers Trainable Pooling | Bi-directional | **+0.0166*** | **-0.0143*** | **+0.0226*** | **-0.0246*** |

**Finding (4): There is no one-size-fits-all winner.** The results in Table 4 demonstrate the varying efficacy of different pooling and attention strategies across multiple tasks. For the STS and retrieval tasks, Multi-Layers Trainable Pooling + Bidirectional (Model 5) significantly and substantially outperforms the other models. For example, Model 5 achieves a 4.2% improvement (+0.0226) over Model 1, which is a standard setting in training LLM-based embedding models, on the retrieval task. However, the Model 5 configuration is less effective for the classification and clustering task, where a more directed, causal attention strategy performs significantly higher. Therefore, there is no one-size-fits-all solution, and the effectiveness of pooling and attention strategies appears to be task-dependent. That being said, we believe that the superiority in STS and retrieval tasks suggests the Multi-Layers Trainable Pooling + Bidirectional (Model 5) might be a more viable choice for practitioners, given that LLM-based embeddings are "essential building blocks for semantic search and retrieval-augmented generation (RAG), which is the predominant approach for domain-specific or company-specific chatbots and other AI application"[3].

# 6 Robustness Check: Qwen as the base LLM

To confirm the robustness of our analysis efficiently, we further fine-tune Model 1 ∼ Model 5 based on Qwen2-0.5B (Yang et al., 2024) using the same training data and evaluate their performance on the MTEB benchmark. The results are presented in Table 5. The reason for choosing this smaller base model is that an embedding model based on a smaller LLM might be more suitable and inference-efficient in resource-constrained situations, and training a smaller LLM-based embedding model is practically relevant (Li et al., 2023). The findings are largely consistent.

First, the overall performance across the four tasks is lower than that reported in Table 4, indicating a significant capacity gap between the base LLMs, Mistral-7B and Qwen2-0.5B. Second, similarly Multi-Layers Trainable Pooling + Bidirectional (Model 5) significantly outperforms the other models on the STS but performs significantly worse on the classification task. This result aligns with Finding 2 and Finding 3, demonstrating that Multi-Layers Trainable Pooling and Bidirectional attention are capable of encoding more contextual information, which is advantageous for retrieval and STS tasks. However, for classification tasks, such as news article classification, the directionality of a text sequence may be less critical, as global semantics play a more crucial role in classification. Third, except for the STS task, the simple EOS-Last token pooling with causal attention (Model 1) performs on par with the trainable pooling methods and bidirectional attention. This suggests that for embedding models based on a smaller LLM, employing more complex designs in pooling and attention strategies does not yield meaningful gains. Moreover, the mixed results underscore the complexity of benchmarking an embedding model and advocate for researchers to conduct statistical significance tests when reporting results.

---

[3]VoyageAI. (September 4, 2024 version), https://docs.voyageai.com/docs/introduction

Table 5: Comparison of different pooling and attention combinations on MTEB benchmark. **Qwen2-0.5B** is the base LLM. The row in `gray` is the baseline for comparison, and the pairwise significant results are marked with asterisks*.

| Combination | Pooling | Attention | STS | Clas. | Retr. | Clus. |
|---|---|---|---|---|---|---|
| Model 1 | EOS Token Pooling | Casual | 0.7765 | 0.6903 | 0.3867 | 0.3885 |
| Model 2 | Last-Layer Trainable Pooling | Casual | +0.0250* | -0.0183 | -0.0280 | -0.0078 |
| Model 3 | Multi-Layers Trainable Pooling | Casual | +0.0268* | -0.0347 | -0.0084 | -0.0035 |
| Model 4 | Last-Layer Trainable Pooling | Bi-directional | +0.0234* | -0.0333* | -0.0013 | +0.0103 |
| Model 5 | Multi-Layers Trainable Pooling | Bi-directional | +0.0372* | -0.0393* | +0.0003 | +0.0019 |

## 7 RELATED WORKS

### 7.1 ENCODER-BASED EMBEDDING MODELS

Text embedding has evolved with the Transformer architecture (Vaswani et al., 2017). Encoder models, particularly BERT (Devlin et al., 2019) and the T5 Encoder (Raffel et al., 2020), have been widely used in tasks like text similarity by capturing sentence-level semantics. Building upon this foundation, Sentence-BERT (Reimers, 2019) uses Siamese networks for fixed-size embeddings to efficiently retrieve semantically similar sentences. Furthermore, the INSTRUCTOR model (Su et al., 2023) leverages the T5 Encoder to incorporate various instructional prompts, allowing it to adapt to a wide range of downstream tasks. The BGE-M3 model (Chen et al., 2024), based on XLM-RoBERTa (Conneau et al., 2020), integrates dense and sparse retrieval to support multi-granularity in the retrieval process.

### 7.2 LLM-BASED EMBEDDING MODELS

The success of LLMs in text generation tasks has sparked increasing interest in exploring LLM-based embedding models. RepLLama (Ma et al., 2024) pioneered this promising direction by finetuning an LLM as a dense retriever, demonstrating the potential of LLMs in embedding tasks. Building upon this foundation, subsequent research has explored various techniques to enhance LLM-based embedding models. E5-mistral-7b-instruct (Wang et al., 2023) investigated the usage of synthetic data in the training process. Recognizing the significance of capturing bidirectional context in embedding tasks, GritLM (Muennighoff et al., 2024) and LLM2Vec (BehnamGhader et al., 2024) employ bidirectional attention mechanisms in their LLM architectures. By attending to both past and future tokens, these models can generate more contextually informed embeddings, potentially leading to improved performance in tasks such as text similarity and retrieval. Furthermore, NV-Embed (Lee et al., 2024) introduces a novel latent attention layer to obtain pooled embeddings for a sequence of tokens. These advancements showcase the ongoing efforts to harness the power of LLMs for embedding tasks. However, existing LLM-based embedding models are often trained on different datasets, leading to mixed conclusions regarding the effectiveness of pooling and attention strategies. Our work aims to empirically evaluate and deepen our understanding of the training design choices for LLM-based embeddings.

## 8 CONCLUSION

In this study, we investigate LLM-based embedding models, focusing on two key design elements: pooling and attention. We conduct a large-scale experiment by fine-tuning five LLM-based embedding models on the same training data using different pooling and attention strategies. Our findings highlight that fine-tuning LLMs with bidirectional attention and an additional trainable pooling layer demonstrates superior performance in semantic textual similarity and information retrieval tasks, but underperforms in clustering and classification tasks. Furthermore, we introduce a new and effective pooling method, Multi-Layers Trainable Pooling, which leverages all layers rather than just the last hidden layer to capture broader and potentially more relevant semantic information. We hope this work sheds light on training LLM-based embedding models.

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

# A   TRAINING AND BENCHMARK DETAILS

The training datasets for LLM-based embedding models are listed in Table 6, and evaluation instructions are listed in Table 7. We use the same instructions as in (Wang et al., 2023) to facilitate easy replication.

Table 6: Training Dataset Overview and Instructions

| Dataset | Examples | Instruction |
|---|---|---|
| STSB (Cer et al., 2017) | 937 | Retrieve semantically similar text. |
| MSMARCO document (Bajaj et al., 2016) | 73, 400 | Given a web search query, retrieve relevant documents that answer the query. |
| Quora Duplicates (DataCanary et al., 2017) | 101, 762 | Given a question, retrieve questions that are semantically equivalent to the given question. |
| MSMARCO passage (Bajaj et al., 2016) | 249, 592 | Given a web search query, retrieve relevant passages that answer the query. |
| NQ (Kwiatkowski et al., 2019) | 100, 231 | Given a question, retrieve Wikipedia passages that answer the question. |
| SQuAD (Rajpurkar et al., 2018) | 87, 599 | Retrieve Wikipedia passages that answer the question. |
| TriviaQA (Joshi et al., 2017) | 73, 346 | Retrieve Wikipedia passages that answer the question. |
| AllNLI (Bowman et al., 2015) | 277, 230 | Given a premise, retrieve a hypothesis that is entailed by the premise. |
| finance-alpaca (gbharti, 2023) | 35, 038 | Given a finance question, retrieve passages that answer the question. |
| FiQA (Maia et al., 2018) | 7, 203 | Given a finance question, retrieve passages that answer the question. |
| MIRACL (Zhang et al., 2022) | 32, 561 | Given a question, retrieve passages that answer the question. |
| xsum (Narayan et al., 2018) | 24, 626 | Given a document, retrieve semantically similar summaries. |
| Mr. TYDI (Zhang et al., 2021) | 48, 715 | Given a question, retrieve Wikipedia passages that answer the question. |
| Altlex (Hidey and McKeown, 2016) | 54, 674 | Retrieve semantically similar text. |
| HotpotQA (Yang et al., 2018) | 90, 447 | Given a multi-hop question, retrieve documents that can help answer the question. |
| ELI5 (Fan et al., 2019) | 32, 547 | Provided a user question, retrieve the highest voted answers on Reddit ELI5 forum. |
| FEVER (Thorne et al., 2018) | 101, 578 | Given a claim, retrieve documents that support or refute the claim. |
| PubMedQA (Jin et al., 2019) | 500 | Given a biomedical question, retrieve information that answers the question. |

# B   DETAILED MTEB EVALUATION RESULTS

The detailed MTEB (Muennighoff et al., 2022) evaluation results for each task across different embedding models are illustrated in the tables 8 to 13. The reported metrics represent the main scores for each task as defined by MTEB. Specifically, we report the Spearman correlation of cosine similarity for the Semantic Textual Similarity (STS) task. We utilize the Normalized Discounted Cumulative Gain (NDCG) at rank 10 for the Retrieval task. For the Classification task, we report Accuracy. Lastly, for the Clustering task, the V-measure metric is used.

Table 7: Evaluation Instructions for MTEB Benchmark

| Dataset | Instruction |
|---|---|
| NFCorpus | Given a question, retrieve relevant documents that best answer the question. |
| ArguAna | Given a claim, find documents that refute the claim. |
| ClimateFEVER | Given a claim about climate change, retrieve documents that support or refute the claim. |
| DBPedia | Given a query, retrieve relevant entity descriptions from DBPedia. |
| FEVER | Given a claim, retrieve documents that support or refute the claim. |
| FiQA2018 | Given a financial question, retrieve user replies that best answer the question. |
| HotpotQA | Given a multi-hop question, retrieve documents that can help answer the question. |
| MSMARCO | Given a web search query, retrieve relevant passages that answer the query. |
| NQ | Given a question, retrieve Wikipedia passages that answer the question. |
| QuoraRetrieval | Given a question, retrieve questions that are semantically equivalent to the given question. |
| SCIDOCS | Given a scientific paper title, retrieve paper abstracts that are cited by the given paper. |
| SciFact | Given a scientific claim, retrieve documents that support or refute the claim. |
| Touche2020 | Given a question, retrieve detailed and persuasive arguments that answer the question. |
| TRECCOVID | Given a query, retrieve documents that answer the query. |
| FinanceBench | Given a financial question, retrieve user replies that best answer the question. |
| Company2Industry | Given a company name, retrieve the related industry. |
| BIOSSES | Retrieve semantically similar text. |
| SICK-R | Retrieve semantically similar text. |
| STS12 | Retrieve semantically similar text. |
| STS13 | Retrieve semantically similar text. |
| STS14 | Retrieve semantically similar text. |
| STS15 | Retrieve semantically similar text. |
| STS16 | Retrieve semantically similar text. |
| STSBenchmark | Retrieve semantically similar text. |
| AskUbuntuDupQuestions | Retrieve duplicate questions from AskUbuntu forum. |
| MindSmallReranking | Retrieve relevant news articles based on user browsing history. |
| SciDocsRR | Given a title of a scientific paper, retrieve the titles of other relevant papers. |
| StackOverflowDupQuestions | Retrieve duplicate questions from StackOverflow forum. |
| AmazonPolarityClassification | Classify Amazon reviews into positive or negative sentiment. |
| ToxicConversationsClassification | Classify if the given comments as either toxic or not toxic. |
| Banking77Classification | Given an online banking query, find the corresponding intents. |
| EmotionClassification | Classify the emotion expressed in the given Twitter message into one of the six emotions: anger, fear, joy, love, sadness, and surprise. |
| ImdbClassification | Classify the sentiment expressed in the given movie review text from the IMDB dataset. |
| TweetSentimentExtractionClassification | Classify the sentiment of a given tweet as either positive, negative, or neutral. |
| SummEval | Given a news summary, retrieve other semantically similar summaries. |
| TwentyNewsgroupsClustering | Identify the topic or theme of the given news articles. |
| ArxivClusteringP2P | Identify the main and secondary category of Arxiv papers based on the titles and abstracts. |
| ArxivClusteringS2S | Identify the main and secondary category of Arxiv papers based on the titles. |
| BiorxivClusteringP2P.v2 | Identify the main category of Biorxiv papers based on the titles and abstracts. |
| BiorxivClusteringS2S.v2 | Identify the main category of Biorxiv papers based on the titles. |
| MedrxivClusteringP2P.v2 | Identify the main category of Medrxiv papers based on the titles and abstracts. |
| MedrxivClusteringS2S.v2 | Identify the main category of Medrxiv papers based on the titles. |
| RedditClustering.v2 | Identify the topic or theme of Reddit posts based on the titles. |
| RedditClusteringP2P.v2 | Identify the topic or theme of Reddit posts based on the titles and posts. |
| StackExchangeClustering.v2 | Identify the topic or theme of StackExchange posts based on the titles. |
| StackExchangeClusteringP2P.v2 | Identify the topic or theme of StackExchange posts based on the given paragraphs. |
| TwentyNewsgroupsClustering.v2 | Identify the topic or theme of the given news articles. |
| TwitterURLCorpus | Retrieve tweets that are semantically similar to the given tweet. |
| SprintDuplicateQuestions | Retrieve duplicate questions from Sprint forum. |
| TwitterSemEval2015 | Retrieve tweets that are semantically similar to the given tweet. |

Table 8: MTEB results on STS tasks.

| Combination | BaseModel | Avg. | BIOSSES | SICK-R | STS12 | STS13 | STS14 | STS15 | STS16 | STSBenchmark |
|---|---|---|---|---|---|---|---|---|---|---|
| Model 1 | Mistral-7B | 0.8302 | 0.8530 | 0.8255 | 0.7147 | 0.8464 | 0.8055 | 0.8777 | 0.8529 | 0.8659 |
| Model 2 | Mistral-7B | 0.8431 | 0.8699 | 0.8309 | 0.7424 | 0.8553 | 0.8227 | 0.8858 | 0.8593 | 0.8786 |
| Model 3 | Mistral-7B | 0.8420 | 0.8641 | 0.8316 | 0.7410 | 0.8589 | 0.8195 | 0.8839 | 0.8596 | 0.8777 |
| Model 4 | Mistral-7B | 0.8397 | 0.8650 | 0.8392 | 0.7368 | 0.8447 | 0.8200 | 0.8811 | 0.8547 | 0.8763 |
| Model 5 | Mistral-7B | 0.8468 | 0.8808 | 0.8389 | 0.7441 | 0.8572 | 0.8322 | 0.8844 | 0.8585 | 0.8782 |

Table 9: MTEB results on Retrieval tasks: Part 1.

| Combination | Model | Avg. | ArguAna | ClimateFEVER | DBPedia | FEVER | FiQA2018 | HotpotQA | MSMARCO |
|---|---|---|---|---|---|---|---|---|---|
| Model 1 | Mistral-7B | 0.5394 | 0.4863 | 0.3814 | 0.4828 | 0.9076 | 0.5262 | 0.6567 | 0.4158 |
| Model 2 | Mistral-7B | 0.5529 | 0.5408 | 0.4009 | 0.4579 | 0.9109 | 0.5582 | 0.6493 | 0.4175 |
| Model 3 | Mistral-7B | 0.5496 | 0.5052 | 0.4206 | 0.4524 | 0.9119 | 0.5553 | 0.6689 | 0.4198 |
| Model 4 | Mistral-7B | 0.5607 | 0.5489 | 0.3772 | 0.4502 | 0.9177 | 0.5888 | 0.6891 | 0.4319 |
| Model 5 | Mistral-7B | 0.5620 | 0.5551 | 0.4024 | 0.4432 | 0.9150 | 0.5747 | 0.6852 | 0.4261 |

Table 10: MTEB results on Retrieval tasks: Part 2.

| Combination | Model | NFCorpus | NQ | QuoraRetrieval | SCIDOCS | SciFact | Touche2020 | TRECCOVID |
|---|---|---|---|---|---|---|---|---|
| Model 1 | Mistral-7B | 0.3609 | 0.6433 | 0.8894 | 0.1903 | 0.7582 | 0.2359 | 0.6175 |
| Model 2 | Mistral-7B | 0.3748 | 0.6413 | 0.8910 | 0.1881 | 0.7332 | 0.2381 | 0.6938 |
| Model 3 | Mistral-7B | 0.3699 | 0.6429 | 0.8900 | 0.1889 | 0.7474 | 0.2344 | 0.7322 |
| Model 4 | Mistral-7B | 0.3901 | 0.6543 | 0.8931 | 0.2023 | 0.7418 | 0.2348 | 0.7291 |
| Model 5 | Mistral-7B | 0.3808 | 0.6636 | 0.8923 | 0.1980 | 0.7403 | 0.2414 | 0.7505 |

Table 11: MTEB results on Classification tasks.

| Combination | Model | Banking77 | Emotion | TweetSentiment | Amazon Polarity | Toxic Conversations | Imdb |
|---|---|---|---|---|---|---|---|
| Model 1 | Mistral-7B | 0.8269 | 0.4871 | 0.6044 | 0.9161 | 0.6342 | 0.8777 |
| Model 2 | Mistral-7B | 0.8319 | 0.4610 | 0.6048 | 0.9161 | 0.6340 | 0.8777 |
| Model 3 | Mistral-7B | 0.8353 | 0.4875 | 0.6029 | 0.9108 | 0.6249 | 0.8650 |
| Model 4 | Mistral-7B | 0.8345 | 0.4638 | 0.5773 | 0.8647 | 0.5973 | 0.7189 |
| Model 5 | Mistral-7B | 0.8304 | 0.5037 | 0.6067 | 0.9069 | 0.6275 | 0.7853 |

Table 12: MTEB results on Clustering tasks: Part 1.

| Combination | Model | ArxivClusteringP2P | ArxivClusteringS2S | BiorxivClusteringP2P | BiorxivClusteringS2S | MedrxivClusteringP2P | MedrxivClusteringS2S |
|---|---|---|---|---|---|---|---|
| Model 1 | Mistral-7B | 0.4896 | 0.4462 | 0.3840 | 0.3686 | 0.3334 | 0.3248 |
| Model 2 | Mistral-7B | 0.4842 | 0.4469 | 0.3838 | 0.3756 | 0.3356 | 0.3339 |
| Model 3 | Mistral-7B | 0.4844 | 0.4559 | 0.3823 | 0.3765 | 0.3398 | 0.3436 |
| Model 4 | Mistral-7B | 0.4419 | 0.4212 | 0.3457 | 0.3289 | 0.3309 | 0.3068 |
| Model 5 | Mistral-7B | 0.4653 | 0.4314 | 0.3549 | 0.3534 | 0.3366 | 0.3291 |

Table 13: MTEB results on Clustering tasks: Part 2.

| Combination | Model | RedditClusteringP2P | StackExchangeClustering | StackExchangeClusteringP2P | TwentyNewsgroupsClustering | RedditClustering |
|---|---|---|---|---|---|---|
| Model 1 | Mistral-7B | 0.6408 | 0.5569 | 0.3982 | 0.4931 | 0.5182 |
| Model 2 | Mistral-7B | 0.6389 | 0.4972 | 0.4155 | 0.4461 | 0.5114 |
| Model 3 | Mistral-7B | 0.6381 | 0.5003 | 0.4099 | 0.4718 | 0.5324 |
| Model 4 | Mistral-7B | 0.5782 | 0.3944 | 0.3861 | 0.4329 | 0.4441 |
| Model 5 | Mistral-7B | 0.6190 | 0.4516 | 0.3966 | 0.4455 | 0.4997 |

