# OpenReview forum: "Pooling And Attention: What Are Effective Designs For LLM-Based Embedding Models?"
_ICLR.cc/2025/Conference — Submitted to ICLR 2025_

### Official Review · Reviewer_jm3o · 2024-10-31

**Soundness:** 2
**Presentation:** 4
**Contribution:** 2
**Rating:** 5
**Confidence:** 3

**Summary:**

Existing LLM-based embedding models employ various pooling methods (such as EOS-last token pooling, mean pooling, and trainable pooling layers) and attention mechanisms (causal vs. bidirectional), but are often trained on different datasets and base models, making it difficult to isolate the impact of these design choices.

To address this, the authors conduct a large-scale experiment where they fine-tune a series of LLM-based embedding models using the same training data and base model (Mistral-7B and Qwen2-0.5B), varying only the pooling and attention strategies. They evaluate these models on the Massive Text Embedding Benchmark (MTEB) and use statistical significance testing to assess the results.

They also propose a new pooling strategy called Multi-Layers Trainable Pooling, which leverages hidden states from all layers of the LLM (not just the last layer) and uses a cross-attention network to produce the final embedding. This method proves to be statistically superior in STS and retrieval tasks compared to existing pooling methods.

**Strengths:**

## Originality:
1. The paper presents a systematic and controlled study that isolates the impact of pooling and attention strategies on LLM-based embedding models. By training multiple models using the same base LLM and training data but varying only the pooling and attention mechanisms, the authors offer new insights into how these factors affect performance across different tasks.
2. Moreover, the introduction of multi-layers trainable pooling strategy based on hidden states from all layers of the LLM, not just the last layer, is novel. This approach aims to capture complementary information encoded at different layers

## Quality:
1. The authors ensure a fair comparison by using the same base models (Mistral-7B and Qwen2-0.5B) and training data across all experiments. This eliminates confounding variables and provide a nice study.
2. Employing the Wilcoxon Signed Rank Test to assess the results adds rigor to the evaluation, providing confidence that the observed differences are statistically meaningful rather than due to random chance.
3. The models are evaluated on the Massive Text Embedding Benchmark (MTEB), covering a wide range of tasks such as semantic textual similarity, retrieval, classification, and clustering.

## Clarity:
1. I like how the paper is introducing notations and concepts in a gradual and comprehensible manner. Complex ideas are broken down and explained step by step, which helps in understanding.
2. Moreover, authors took care to use consistent notation throughout the paper, which helps prevent confusion and allows to follow the methodology and results more easily.
3. Finally, figures, tables, and diagrams are used to illustrate key points, such as the correlation between hidden states across layers and the architecture of the proposed pooling method.

## Significance:

1. Practical Insights - by revealing that there is no one-size-fits-all solution for pooling and attention strategies in LLM-based embedding models, the paper is valuable for practitioners. The findings suggest that the choice of strategy should be task-dependent.
2. The introduction of the Multi-Layers Trainable Pooling method contributes to the field by proposing a new way to utilize the rich information contained in the various layers of an LLM.

**Weaknesses:**

## Overemphasis on Technical Novelty over Practical Problem Solving

1. The paper focuses on the proposed method over problem understanding. The paper seems to prioritize introducing a novel technical method without fully addressing whether it effectively solves the underlying problem of improving embedding models. The connection between the proposed solution and the practical challenges in embedding generation is not thoroughly established.
2. It lacks a fundamental study on embedding models. The study does not delve deeply into fundamental aspects of what makes a good embedding model for specific tasks. Without this foundational understanding, it's difficult to assess whether the proposed method addresses the core issues in embedding generation. Without this, it seems like we are throwing random methods at the problem without increasing our understanding of what is the root cause and what impacts whether an embedding model is good.


## Insufficient Ablation Studies and Hyperparameter Analysis

1. No ablations on key hyperparameters are provided. The paper lacks ablation studies on important hyperparameters, such as the number of queries trained (r), and the inner dimension d' of the cross-attention block or LoRA rank. Exploring how these parameters affect performance would provide deeper insights into the robustness and effectiveness of the proposed method.

2. It seems there is a gap in models evaluated as comparison with simplified versions is lacking. Investigating simpler versions of the model, such as training the attention matrix (author's way) on only the last layer's output (similar to NV-embed), could help isolate the benefits of incorporating multiple layers. This comparison is essential to justify the added complexity of the Multi-Layers Trainable Pooling. From what I understand - it seems that the last-layer approach is technically different from the multi-layer approach, and thus, not directly comparable.




## Questionable Robustness and Generalizability of Results

1. The performance is mixed across tasks. The proposed method does not consistently outperform simpler baselines across all tasks. While it shows improvements in semantic textual similarity (STS) and retrieval tasks (Table 4), it underperforms in classification and clustering tasks. This inconsistency raises concerns about the robustness and generalizability of the method.
2. There is potential randomness in results. Without more extensive experimentation or replication studies, it's possible that the observed improvements are due to random chance rather than a fundamental advantage of the method. The limited scale of the study may not provide enough empirical evidence to draw firm conclusions. I am worried that simply changing the training horizon will impact the conclusions drawn by authors. The results on Mistral are not convincingly transferred to QWEN and the reason remain unclear. I think the lack of the fundamental study mentioned earlier takes it toll here.

**Questions:**

1. Your method shows improvements in semantic textual similarity (STS) and retrieval tasks but underperforms in classification and clustering tasks. Could you provide an analysis or explanation for this inconsistency? Is there an underlying reason why the proposed method benefits some tasks but not others? Understanding this could help practitioners decide when to apply your method.
2. Your findings suggest that we may not fully grasp how to optimize embedding models, and further exploration of this topic could provide valuable insights for the community.
3. To ensure that the observed improvements are not due to random chance, have you considered conducting experiments with multiple random seeds or on additional datasets? Providing more extensive empirical evidence would strengthen the validity of your conclusions.
4. How does your method compare with simpler variants, such as training the attention matrix on only the last layer's output (similar to NV-embed)? Including such comparisons would help isolate the benefits of incorporating multiple layers in your pooling strategy. An ablation study that progressively adds complexity could demonstrate the necessity of each component.

---

> ### Author Response · Authors · 2024-11-20
>
> Thank you for highlighting the need for deeper analysis of task-specific performance variations. We have conducted additional experiments that provide important insights. Below we address the main concerns:
>
> **1.Explanation About STS/Retrieval vs Classification/Clustering**
>
> Inspired by reviewers' questions about task-specific performance variations, we conducted a controlled experiment to investigate how different training data compositions affect model performance.
>
> **This experiment confirmed our findings in the paper, which suggested that the optimization objectives of STS/Retrieval and Classification/Clustering may not be consistent.**
>
> We compared two settings:
>
> - Baseline: 68,039 samples (random sampled from the original dataset)
> - Enhanced: 136,078 samples (68,039 retrieval + 68,039 classification/clustering tasks)
> - All training data has no overlap with testing data.
>
>
> | Model            | STS   | Retrieval | Classification | Clustering | Average |
> |------------------|-------|-----------|----------------|------------|---------|
> | model1           | 0.6445| 0.2970    | 0.6517         | 0.3965     | 0.4974  |
> | model1-enhanced  | 0.6446| 0.2398    | 0.6048         | 0.4147     | 0.4759  |
> | model2           | 0.7301| 0.3454    | 0.6983         | 0.4049     | 0.5447  |
> | model2-enhanced  | 0.6815| 0.3068    | 0.6779         | 0.4253     | 0.5229  |
> | model3           | 0.7165| 0.3407    | 0.6999         | 0.4082     | 0.5413  |
> | model3-enhanced  | 0.6581| 0.2856    | 0.6797         | 0.4240     | 0.5119  |
> | model4           | 0.7601| 0.3673    | 0.7058         | 0.4263     | 0.5649  |
> | model4-enhanced  | 0.7040| 0.3150    | 0.6851         | 0.4480     | 0.5380  |
> | model5           | 0.7700| 0.3708    | 0.7100         | 0.4505     | 0.5753  |
> | model5-enhanced  | 0.7200| 0.3310    | 0.6920         | 0.4700     | 0.5532  |
>
> This suggests that:
>
> 1. Different tasks have potentially conflicting optimization objectives
> 2. Performance trade-offs are inherent, not architectural limitations
> 3. Task-specific training data is crucial for optimal performance
>
>
> **2. Additional Datasets**
>
> We appreciate your feedback on the robustness of work. We evaluated our **original models** in the paper on additional datasets to ensure their generalizability. These additional datasets further confirm our findings, demonstrating the robustness and generalizability of our method.
>
> The results are summarized below:
>
> | Task / Metric | STS / cos_sim spearman | Retrieval / ndcg_at_10 | Classification / Accuracy | Clustering / Validity Measure (V-measure) |
> | --- | --- | --- | --- | --- |
> | **Model 1** | 0.3689 | 0.6066 | 0.6640 | 0.7607 |
> | **Model 2** | 0.3534 | 0.6152 | 0.6640 | 0.7823 |
> | **Model 3** | 0.3567 | 0.6202 | 0.6519 | 0.8018 |
> | **Model 4** | 0.3786 | 0.5998 | 0.5634 | 0.6212 |
> | **Model 5** | 0.3864 | 0.6070 | 0.6426 | 0.7744 |
>
>
> STS Datasets: FINAL (Ju et al., 2023), FinSTS (Liu et al., 2024)
>
> Retrieval Datasets: FiQA2018 (FiQA, 2018), FinanceBench  (Islam et al.,2023), HC3 (Guo et al., 2023), Apple10KRetrieval (Tang et al., 2024), FinQA (Chen et al., 2021)
>
> Classification Datasets: FinancialPhrasebank (Malo et al., 2014), FinSent (Yang et al., 2023), FiQA, SemEva2017 Headline (Cortis et al., 2017), FLS (Yang et al., 2023), ESG (Yang et al., 2023), FOMC (Shah et al., 2023)
>
> Clustering Datasets: MInDS-14-zh (Gerz et al., 2021), WikiCompany2Industry (Tang et al., 2024)
>
> **3. NV-embed Style Model**
>
> We would like to clarify that Model 4 in our paper serves as an NV-Embedded-style trainable last-layer pooling with a slight difference: While NV-Embed transforms the last layer's hidden states to the query matrix Q, our method transforms it into K matrix and V matrix for apple-to-apple comparison with Model 5 (Add trainable layer weights).
>
> For the rigorous evaluation, we also implemented a NV-Embed model using the dataset from question 1. The results are illustrated in the table:
>
> | Model            | STS   | Retrieval | Classification | Clustering | Average |
> |------------------|-------|-----------|----------------|------------|---------|
> | model1           | 0.6445| 0.2970    | 0.6517         | 0.3965     | 0.4974  |
> | model2           | 0.7301| 0.3454    | 0.6983         | 0.4049     | 0.5447  |
> | model3           | 0.7165| 0.3407    | 0.6999         | 0.4082     | 0.5413  |
> | model4(NV-Embed style)         | 0.7601| 0.3673    | 0.7058         | 0.4263     | 0.5649  |
> | model5           | 0.7700| 0.3708    | 0.7100         | 0.4505     | 0.5753  |
> | model6(Same with NV-Embed) | 0.6992 |   0.3426    | 0.6804           | 0.4136      |  0.5340   |
>
> The performance of the NV-Embed model is also consistent with the findings in the paper and did not surpass the model 5 (with layer weights).
>
> Thank you again for your valuable feedback. If you have any other questions or require more clarification, please do not hesitate to let us know!

---

> > ### Author Response · Authors · 2024-11-26
> >
> > Dear Reviewer,
> >
> > Thank you for handling our manuscript and providing valuable feedback. We hope that our responses have sufficiently addressed the concerns you raised. We welcome more discussion if you have more questions and suggestions. As the discussion deadline is approaching, we would be very grateful if you could take a moment to review our reply.
> >
> > Best，
> >
> > The Authors

---

### Official Review · Reviewer_YfBn · 2024-11-03

**Soundness:** 2
**Presentation:** 3
**Contribution:** 2
**Rating:** 3
**Confidence:** 4

**Summary:**

This paper studies the design space of attention and pooling for dense retriever models that are finetuned from an LLM. The paper studies bidirectional and causal attention masking. It also explores mean, last-token and trainable pooling types. For the trainable pooling types it studies the use of only last-layer representations and multiple-layer representations. The empirical results show that there is no one-size-fits-all solution and that various tasks have different optimal designs.

**Strengths:**

1) The work introduces the novel concept of pooling across layer representations for LLM based dense retrieval models.
2) The work constructs and evaluates an architecture for multi-layer trainable pooling and it performs better than other pooling types in certain scenarios.
3) The work is well motivated: various related works cover various portions of the design space but they use different datasets, so it is important to study the design space while keeping the dataset constant.

**Weaknesses:**

This paper seems to be a rigorous study of the design space of dense embedding models rather than an attempt at the state-of-the-art, However, the experiments are not rigorous enough:
1) The evaluations (Tables 2-4) and discussion (Section 5) include classification and retrieval tasks. However, the training datasets (Table 6) do not include either of these tasks.
2) Bidirectional attention with mean-pooling is not evaluated with the reasoning in Section 4.1 that NV-Embed, a related work has shown that trainable pooling can outperform mean pooling. However, since the trainable last-layer pooling in this paper is not the same as in NV-Embed, this study should have included bidirectional attention with mean-pooling and NV-Embed style trainable last-layer pooling as baselines.
3) In section 3.1, experiment 2, it is observed that the EOS token from layer 0 of Mistral performs significantly better in retrieval than all later layers, which have nearly 0 scores. This seems unlikely since at layer 0, the EOS token is unlikely to have a strong representation of the text. Furthermore, why is EOS pooling used for this experiment when LLM2Vec has shown that mean pooling outperforms EOS pooling even for the causal Mistral model without any finetuning.

**Questions:**

1) If the importance of various layers is task-dependent, would the model perform better if trainable Q latent matrix was task dependent? One idea would be to produce the Q from a representation of the task instruction. Another experiment towards that idea would be to have a separate Q matrix for STS, Retrieval, Clustering, Classification and see if those improved results.
2) Why is the layer weight matrix needed if the cross-attention block is anyway weighing the layer representation using attention weights? What are the results if the layer weight matrix was not used?

Further ablations and datasets related to the weaknesses above would improve the rigor of the paper:
1) Include classification and clustering in the training data or exclude any conclusions for classification and clustering.
2) Compare with mean pooling+bidirectional attention and NV-Embed style last-layer trainable pooling.

---

> ### Author Response · Authors · 2024-11-20
>
> Thank you for your thorough review. We appreciate your recognition of our work's strengths and your constructive feedback. Let us address each of your concerns:
>
>
> **1. Missing Training Data of Classification/Clustering Tasks**
>
> Regarding our training dataset selection, we initially aligned with current standard training pipelines (such as LLM2Vec [1]) to ensure fair comparison. However, your question raised an important point about potential task-specific biases.
>
> Inspired by your suggestion and other reviewers' feedback, we conducted a new controlled experiment to investigate how training data composition affects different downstream tasks.
>
> **Key findings** show that while balanced training data improved clustering performance, it led to slight decreases in retrieval/STS tasks. This trade-off suggests different tasks may indeed require different optimization objectives, aligning with our paper's findings.
>
> We compared two settings:
>
> - Baseline: 68,039 samples (random sampled from the original dataset)
> - Enhanced: 136,078 samples (68,039 retrieval + 68,039 classification/clustering tasks)
> - All training data has no overlap with testing data.
>
>
> | Model            | STS   | Retrieval | Classification | Clustering | Average |
> |------------------|-------|-----------|----------------|------------|---------|
> | model1           | 0.6445| 0.2970    | 0.6517         | 0.3965     | 0.4974  |
> | model1-enhanced  | 0.6446| 0.2398    | 0.6048         | 0.4147     | 0.4759  |
> | model2           | 0.7301| 0.3454    | 0.6983         | 0.4049     | 0.5447  |
> | model2-enhanced  | 0.6815| 0.3068    | 0.6779         | 0.4253     | 0.5229  |
> | model3           | 0.7165| 0.3407    | 0.6999         | 0.4082     | 0.5413  |
> | model3-enhanced  | 0.6581| 0.2856    | 0.6797         | 0.4240     | 0.5119  |
> | model4           | 0.7601| 0.3673    | 0.7058         | 0.4263     | 0.5649  |
> | model4-enhanced  | 0.7040| 0.3150    | 0.6851         | 0.4480     | 0.5380  |
> | model5           | 0.7700| 0.3708    | 0.7100         | 0.4505     | 0.5753  |
> | model5-enhanced  | 0.7200| 0.3310    | 0.6920         | 0.4700     | 0.5532  |
>
>
> **2. Task Dependent Trainable Q**
>
> Following your suggestion, we implemented a task-dependent Q by concatenating task instruction embeddings with the trainable Q matrix. The experimental setup remained consistent with Model 5, trained on the enhanced dataset. Results below show that this approach led to performance degradation across all metrics:
>
> | Model            | STS   | Retrieval | Classification | Clustering | Average |
> |------------------|-------|-----------|----------------|------------|---------|
> | Model 5 (baseline)| 0.7200| 0.3310    | 0.6920         | 0.4700     | 0.5532  |
> | Task-dependent Q | 0.5772| 0.1455    | 0.5693         | 0.4113     | 0.4258  |
> | Δ                | -0.1428| -0.1855   | -0.1227        | -0.0587    | -0.1274 |
>
> While this simple instruction-based adaptation did not yield improvements, we believe your suggestion opens up promising future directions! We will explore more approaches in this field.
>
> **3. Compare with NV-Embed Style Model**
>
> We would like to clarify that Model 4 in our paper serves as an NV-Embedded-style trainable last-layer pooling with a slight difference: While NV-Embed transforms the last layer's hidden states to the query matrix Q, our method transforms it into K matrix and V matrix for apple-to-apple comparison with Model 5 (Add trainable layer weights).
>
> For the rigorous evaluation, we also implemented a complete NV-Embed model using the dataset from question 1. The results are illustrated in the table:
>
> | Model            | STS   | Retrieval | Classification | Clustering | Average |
> |------------------|-------|-----------|----------------|------------|---------|
> | model1           | 0.6445| 0.2970    | 0.6517         | 0.3965     | 0.4974  |
> | model2           | 0.7301| 0.3454    | 0.6983         | 0.4049     | 0.5447  |
> | model3           | 0.7165| 0.3407    | 0.6999         | 0.4082     | 0.5413  |
> | model4(NV-Embed style)         | 0.7601| 0.3673    | 0.7058         | 0.4263     | 0.5649  |
> | model5           | 0.7700| 0.3708    | 0.7100         | 0.4505     | 0.5753  |
> | model6(Same with NV-Embed) | 0.6992 |   0.3426    | 0.6804           | 0.4136      |  0.5340   |
>
> The performance of the NV-Embed model is also consistent with the findings in the paper and did not surpass the model 5 (with layer weights).

---

> > ### Author Response · Authors · 2024-11-20
> >
> > **4. Layer 1 EOS Token Performance**
> >
> > There appears to be a misunderstanding - the peak performance was observed at layer 1, not layer 0 as you mentioned. The results are empirically derived, and we note that this pattern (layer 1 outperformance) is specific to Mistral - the results in Llama do not show similar outperformance in layer 1. This makes more intuitive sense since, as you pointed out, at layer 0 the EOS token would be unlikely to have a strong representation of the text. Layer 1, having gone through one round of processing, would be better positioned to capture meaningful representations.
> >
> > **5. Why EOS Pooling As Baseline**
> >
> > We chose EOS pooling as our baseline because:
> >
> > 1. It represents the simplest initial approach to finetune an LLM-based embedding model
> > 2. Recent research [1] has shown that mean pooling with causal models can introduce bias for the earlier tokens, leading to decreased performance in embedding tasks.
> >
> > **6. Why is Layer Weights Needed**
> >
> > Layer weights are not weighted by attention weights. Instead, layer weights are added to the hidden layer features to incorporate positional information into the input features. While the attention mechanism can capture relationships between layer features globally, without the positional information from layer weights, it cannot understand the relative positions between layers, which can negatively impact the model's performance.
> >
> > We will incorporate these additional experiments and insights into the final version. Thank you for helping us strengthen the paper's empirical validation. If you have any other questions or require more clarification, please do not hesitate to let us know!
> >
> > ---
> > [1] BehnamGhader, Parishad, et al. 2024. LLM2Vec: Large Language Models Are Secretly Powerful Text Encoders. arXiv preprint arXiv:2404.05961.
> >
> > [2] Jacob Mitchell Springer, Suhas Kotha, Daniel Fried, Graham Neubig, and Aditi Raghunathan. 2024. Repetition improves language model embeddings. arXiv preprint arXiv:2402.15449.

---

> > > ### Comment · Reviewer_YfBn · 2024-11-23
> > >
> > > Thank you for the additional experiments. I appreciate the effort and the experiment results do provide additional insights.
> > > However I continue to maintain my rating for the following reasons:
> > >
> > > >Instead, layer weights are added to the hidden layer features to incorporate positional information into the input features.
> > >
> > > Section 3.2 in the paper mentions:
> > > "we introduce a trainable layer weights matrix that captures the significance of each layer."
> > > Which seem to indicate that the layer weights we designed to weigh the importance of layers rather than differentiate positionally between them. There is no mention of positional encodings, nor an ablation of using positional encodings instead of trainable weights.
> > >
> > >
> > > > Enhanced: 136,078 samples (68,039 retrieval + 68,039 classification/clustering tasks)
> > >
> > > For every enhanced model, the classification score is lower despite adding classification data. This needs further investigation and root-causing before publication. Some suggestions for investigation could be:
> > > 1) Negatives: are there items that could be categorized as other classes? Having those classes as negatives can hurt accuracy.
> > >  2) What if the "documents" for classification are examples of positive and negative class members instead of class labels.
> > >
> > > I appreciate the motivation of this work and look forward to seeing a future publication with more experimental rigor and explanations of experimental results that would serve as a guide for other researchers.

---

> ### Author Response · Authors · 2024-11-23
> **Thanks**
>
> We deeply appreciate the time and effort you have dedicated to reviewing our work.
>
> * **Regarding the statement:**
> ```
> Instead, layer weights are added to the hidden layer features to incorporate positional information into the input features.
> ```
> The positional information here refers to the weights at different layer indices, which is consistent with our statement that "we introduce a trainable layer weights matrix that captures the significance of each layer." The purpose is to find the optimal combination of layers in a trainable way, as the last layer alone may not be optimal for semantic tasks.
>
> * **Regarding the Classification/Clustering Tasks:**
>
>   - Our use of label_text as "documents" follows the experimental setting established in NV-Embed.
>   - Since we use gold-standard labels and treat other labels from the same dataset as negative samples, the question of "whether items could be categorized into other classes" is not applicable in this context.
>
> While we remain confident in our research findings and results, we sincerely appreciate your thoughtful suggestions for deeper exploration. We are excited to continue our investigation in this direction.
>
> Thank you again for your valuable feedback. We have learned a lot from your insights. Wishing you all the best!

---

### Official Review · Reviewer_S4qG · 2024-11-04

**Soundness:** 3
**Presentation:** 3
**Contribution:** 2
**Rating:** 5
**Confidence:** 4

**Summary:**

This paper explores design choices in LLM-based embedding models, focusing on pooling and attention strategies. It fine-tuned different case models using the same dataset but different pooling and attention configurations. The experimental results indicate that bidirectional attention with an additional trainable pooling layer outperforms in STS and retrieval tasks but falls short in clustering and classification tasks. Finally, this study proposes Multi-Layers Trainable Pooling, which utilizes all hidden layers to capture richer semantic information.

**Strengths:**

- The authors identify the key factors (pooling and attention) to transform the decoder-only LLM to embedding models and conducted interesting ablation study.
- The proposed multi-layer trainable pooling is interesting idea, incorporating the semantic information from all layers.

**Weaknesses:**

See the question below.

**Questions:**

- While the paper shows interesting ablation studies, it is not convincing comparing the model with other leading models from MTEB leaderboard. To do that, can you provide the full MTEB evaluation results?
- For training, only retrieval datasets are employed, but other embedding tasks (such as clustering and classification) datasets are not used. Training the model only on retrieval dataset can make the model overfitted to one task to decrease the accuracy of other tasks. This may be the reason why classification and clustering accuracy degrades in model 2 and 3 cases.
- Based on above observation, can you conduct the training adding the clustering and classification datasets for ablation studies?

---

> ### Author Response · Authors · 2024-11-20
>
> We appreciate your recognition of the value of our work in the multi-layer trainable pooling and thoughtful feedback! Below, we address each of your concerns:
>
> **1. Full MTEB Evaluation**
>
> **The primary objective** of our paper is to empirically establish best practices for LLM-based embedding models, rather than to achieve state-of-the-art performance. This focus explains why we didn't directly compare our models with other models in the MTEB benchmark. Instead, we concentrated on four fundamental tasks (STS, retrieval, classification, and clustering) that are most frequently used in real-world applications.
>
> However, for completeness and to address your question, we present the full MTEB evaluation results in the table below:
>
>
> | Model | Pooling | Attention | STS | Classification | Retrieval | Clustering | Pair Classification | Reranking | Summarization | Avg. |
> | --- | --- | --- | --- | --- | --- | --- | --- | --- | --- | --- |
> | Model 1 | EOS-Last Token Pooling | Casual | 0.8302 | 0.7244 | 0.5394 | 0.4503 | 0.8605 | 0.5737 | 0.3240 | 0.6149 |
> | Model 2 | Last-Layer Trainable Pooling | Casual | 0.8431 | 0.7209 | 0.5496 | 0.4427 | 0.8639 | 0.5720 | 0.3097 | 0.6145 |
> | Model 3 | Multi-Layers Trainable Pooling | Casual | 0.8420 | 0.7211 | 0.5529 | 0.4486 | 0.8627 | 0.5787 | 0.2996 | 0.6151 |
> | Model 4 | Last-Layer Trainable Pooling | Bi-directional | 0.8397 | 0.6761 | 0.5607 | 0.4010 | 0.8707 | 0.5829 | 0.3179 | 0.6070 |
> | Model 5 | Multi-Layers Trainable Pooling | Bi-directional | 0.8468 | 0.7101 | 0.5620 | 0.4257 | 0.8746 | 0.5912 | 0.3246 | 0.6193 |
>
> We will also include this table in the Appendix.
>
> **2. STS/Retrieval vs Classification/Clustering Tasks**
>
> Regarding our training dataset selection, we initially aligned with current standard training pipelines (such as LLM2Vec(BehnamGhader et al., 2024)) to ensure fair comparison. However, your question raised an important point about potential task-specific biases.
>
> Inspired by your suggestion and other reviewers' feedback, we conducted a new controlled experiment to investigate how training data composition affects different downstream tasks.
>
> **Key Findings:** This experiment confirmed our findings in the paper, which suggested that the optimization objectives of STS/Retrieval and Classification/Clustering may not be consistent. This explains why a model optimized for one type of task might not naturally excel at the other.
>
>
> We compared two settings:
> - Baseline: 68,039 samples (random sampled from the original dataset)
> - Enhanced: 136,078 samples (68,039 retrieval + 68,039 classification/clustering tasks)
> - All training data has no overlap with testing data.
>
> | Model            | STS   | Retrieval | Classification | Clustering | Average |
> |------------------|-------|-----------|----------------|------------|---------|
> | model1           | 0.6445| 0.2970    | 0.6517         | 0.3965     | 0.4974  |
> | model1-enhanced  | 0.6446| 0.2398    | 0.6048         | 0.4147     | 0.4759  |
> | model2           | 0.7301| 0.3454    | 0.6983         | 0.4049     | 0.5447  |
> | model2-enhanced  | 0.6815| 0.3068    | 0.6779         | 0.4253     | 0.5229  |
> | model3           | 0.7165| 0.3407    | 0.6999         | 0.4082     | 0.5413  |
> | model3-enhanced  | 0.6581| 0.2856    | 0.6797         | 0.4240     | 0.5119  |
> | model4           | 0.7601| 0.3673    | 0.7058         | 0.4263     | 0.5649  |
> | model4-enhanced  | 0.7040| 0.3150    | 0.6851         | 0.4480     | 0.5380  |
> | model5           | 0.7700| 0.3708    | 0.7100         | 0.4505     | 0.5753  |
> | model5-enhanced  | 0.7200| 0.3310    | 0.6920         | 0.4700     | 0.5532  |
>
> We can find:
>
> - Task Balance Impact: Models trained on the enhanced dataset showed improved performance in clustering tasks, supporting your hypothesis about the importance of diverse training data.
>
> - Performance Trade-offs: The observed performance decrease in STS/Retrieval tasks after adding classification/clustering data provides important insights. This suggests that similarity-based tasks and category-based tasks indeed have different, potentially inconsistent optimization objectives, which is consistent with our findings in paper.
>
> To conclude, your suggestions also provide valuable guidance in selecting training data based on their specific application needs. We plan to incorporate these insights into our final paper and remain available for further discussion if you have any additional questions or require more clarification.

---

> ### Comment · Reviewer_S4qG · 2024-11-25
>
> Thank you to the author for conducting additional ablation studies. The *-enhanced model's average scores now align more closely with the ideas of the paper across various embedding benchmark tasks. However, the *-enhanced models perform worse than the baseline retrieval and classification scores. I believe these sub-optimal results can be addressed through improvements in training strategies, dataset blends, hyperparameter tuning, etc. I increase the score of the paper.

---

> > ### Author Response · Authors · 2024-11-26
> > **Thank You!**
> >
> > Dear Reviewer S4qG,
> >
> > Thank you for your careful consideration of our work and insightful comments. We appreciate the willingness to raise the score to 5, which we believe is a positive step toward acceptance. We are happy to address any further questions or concerns!
> >
> > Thank you once again, and we wish you all the best!
> >
> > Best，
> > The Authors

---

### Official Review · Reviewer_pdQZ · 2024-11-04

**Soundness:** 3
**Presentation:** 3
**Contribution:** 2
**Rating:** 5
**Confidence:** 3

**Summary:**

The paper conducts many experiments by training several LLM-based embedding models using the same training data and base model, but varying their pooling and attention strategies. The results indicate that there is no one-size-fits-all solution. Furthermore, the paper proposes a new pooling method called Multi-Layer Trainable Pooling. This method shows improvements in text similarity and retrieval tasks compared to the baseline, but offers no gains in classification and clustering tasks.

**Strengths:**

1. It conducts a large-scale experiment and reports statistical significance.
2. The paper is clearly written and easy to understand.

**Weaknesses:**

1. The paper offers no surprises; it primarily conducts numerous experiments, and the proposed multi-layer trainable pooling method lacks novelty.
2. The improvement is negligible, and the proposed method does not show gains across all tasks.

**Questions:**

Could you explain further why the proposed method works well for text similarity and retrieval tasks, but not for classification and clustering? I believe the underlying reasons might be interesting.

---

> ### Author Response · Authors · 2024-11-20
>
> Thank you for your thoughtful evaluation and constructive feedback on our manuscript. We appreciate the opportunity to clarify our contributions and address your concerns.
>
> **1. Clarification on Contributions&Novelty**
>
> - **Existing Challenge:**
>
> Given that most existing LLM-based embedding models are trained using different datasets with different base models, it is difficult to draw conclusions regarding the contribution of different model architecture design choices (as shown in Table 1).
>
> - **Primary Objective:**
>
> Thus, the primary objective of our paper is to empirically establish best practices for LLM-based embedding models, rather than to achieve state-of-the-art performance. Then, provide insights and guidance on choosing pooling and attention strategies for LLM-based embedding models.
>
>     - Methodology
>       - Conduct controlled experiments using identical base models, training data, and hyperparameters.
>       - Focus on five widely-used model architecture combinations.
>       - Use empirical experiments to ensure that findings are statistically meaningful rather than due to random variation.
>
> - **Secondary Objective**
>
> To better enhance the pooling strategy, we introduced the implementation of multi-layer pooling in LLM-based embedding via cross-attention. This is based on our observation that the last layer hidden state is not always the most semantically relevant layer.
>
> To our knowledge, this is the first method using multi-layer information for LLM-based embedding models.
>
>
> **2. Regarding "Negligible Improvements"**
>
> We respectfully address the concern about "negligible improvements" with two key points:
>
>  - **Context of Existing Works**: As shown in Table, improvements across recent embedding models are typically modest. Even state-of-the-art models achieve only incremental gains, that's why we want to find an optimal model setting using controlled/empirical experiments.
>
> | Model | Avg.|
> |---------|----------------------------------|
> | NV-Embed-v2  (Lee et al.,2024) | 72.31 |
> | bge-en-icl (Li et al.,2024)| 71.67 |
> | dunzhang/stella_en_1.5B_v5| 71.19 |
> | SFR-Embedding-2_R (Meng, et al.,2024) | 70.31 |
> | gte-Qwen2-7B-instruct-GGUF (Li, et al.,2023) | 70.24 |
>
> - **Statistical Validation:** Unlike many previous works, we provide statistical significance tests for all reported improvements, ensuring that our gains, though modest, are reliable and statistically meaningful rather than due to random variation.
>
> **3.Explanation About STS/Retrieval vs Classification/Clustering**
>
> Inspired by reviewers' questions about task-specific performance variations, we conducted a new controlled experiment to investigate how different training data compositions affect model performance.
>
> This experiment confirmed our findings in the paper, which suggested that the optimization objectives of STS/Retrieval and Classification/Clustering may not be consistent. This explains why a model optimized for one type of task might not naturally excel at the other.
>
> - **Methodology**
>
> We compared two settings:
>
> 1. Baseline: 68,039 samples (random sampled from the original dataset).
>
> 2. Enhanced: 136,078 samples (68,039 retrieval + 68,039 classification/clustering tasks)
>
> Our findings reveal that:
>
> 1. With retrieval-only training data (Baseline), models with trainable layers (Model 2-5) significantly outperform EOS-token pooling in similarity-based tasks (STS/Retrieval), but show comparable or slightly worse performance in classification/clustering tasks.
>
> 2. When adding classification/clustering training data (Enhanced), we observe:
>
>     - Performance drops in STS/Retrieval tasks
>     - Improved performance in clustering tasks
>     - Relatively stable classification performance
>
>
> | Model            | STS   | Retrieval | Classification | Clustering | Average |
> |------------------|-------|-----------|----------------|------------|---------|
> | model1           | 0.6445| 0.2970    | 0.6517         | 0.3965     | 0.4974  |
> | model1-enhanced  | 0.6446| 0.2398    | 0.6048         | 0.4147     | 0.4759  |
> | model2           | 0.7301| 0.3454    | 0.6983         | 0.4049     | 0.5447  |
> | model2-enhanced  | 0.6815| 0.3068    | 0.6779         | 0.4253     | 0.5229  |
> | model3           | 0.7165| 0.3407    | 0.6999         | 0.4082     | 0.5413  |
> | model3-enhanced  | 0.6581| 0.2856    | 0.6797         | 0.4240     | 0.5119  |
> | model4           | 0.7601| 0.3673    | 0.7058         | 0.4263     | 0.5649  |
> | model4-enhanced  | 0.7040| 0.3150    | 0.6851         | 0.4480     | 0.5380  |
> | model5           | 0.7700| 0.3708    | 0.7100         | 0.4505     | 0.5753  |
> | model5-enhanced  | 0.7200| 0.3310    | 0.6920         | 0.4700     | 0.5532  |
>
>
> Once again, we sincerely thank you for your valuable feedback and remain available for further discussion if you have any additional questions or require more clarification.

---

> > ### Author Response · Authors · 2024-11-26
> >
> > Dear Reviewer,
> >
> > Thank you for handling our manuscript and providing valuable feedback. We hope that our responses have sufficiently addressed the concerns you raised. We welcome more discussion if you have more questions and suggestions. As the discussion deadline is approaching, we would be very grateful if you could take a moment to review our reply.
> >
> > Best，
> >
> > The Authors

---

> > > ### Comment · Reviewer_pdQZ · 2024-11-26
> > >
> > > Thank you for your additional clarification and experiments. Considering the limitations posed by the lack of surprise and the high demand of the ICLR conference, I will maintain my score of 5.

---

> > > > ### Author Response · Authors · 2024-11-26
> > > > **Thank You!**
> > > >
> > > > Dear Reviewer pdQZ,
> > > >
> > > > We deeply appreciate the time and effort you have dedicated to reviewing our work, as well as your prompt response. Thank you once again, and we wish you all the best!
> > > >
> > > > The Authors

---

### Meta-Review · Area_Chair_TPte · 2024-12-16

**Metareview:**

This submission presents a study of design choices in LLM-based embedding models, focusing on pooling and attention strategies. The paper conducts controlled experiments using identical base models and training data to isolate the impact of different architectural choices. The main contribution is a proposed Multi-Layers Trainable Pooling method that leverages hidden states from all layers. Results show this approach outperforms baselines in STS and retrieval tasks, though performance varies across different task types. While the work offers useful empirical insights, reviewers identified several key concerns: (1) The improvements are relatively modest and inconsistent across tasks, (2) The experimental validation would benefit from more rigorous ablation studies and hyperparameter analysis, (3) The fundamental reasons for task-specific performance variations are not fully explored or explained. Given these limitations and the high bar for NeurIPS, I recommend rejection.

**Additional Comments On Reviewer Discussion:**

The authors' response addressed several reviewer concerns through additional experiments and analysis. They provided more comprehensive MTEB benchmark results and conducted new experiments investigating the impact of training data composition on different tasks. The experiments with enhanced training data (adding classification/clustering samples) revealed interesting trade-offs, suggesting fundamentally different optimization objectives between similarity-based and category-based tasks. The authors also implemented and evaluated an NV-Embed style baseline for comparison. While these additions strengthen the empirical validation, reviewers maintained their concerns about the limited technical novelty and mixed performance across tasks. Reviewer YfBn highlighted the need to further investigate why classification scores decreased despite adding classification training data. Reviewer pdQZ acknowledged the clarifications but maintained their score given the paper's limitations.

---

### Decision · Program_Chairs · 2025-01-22

Reject